# Objective quantitative multidetector computed tomography assessments in patients with combined pulmonary fibrosis with emphysema: Relationship with pulmonary function and clinical events

Masaki Suzuki[1]*, Naoko Kawata[1], Mitsuhiro Abe[1], Hajime Yokota[2], Rie Anazawa[1], Yukiko Matsuura[1], Jun Ikari[1], Shin Matsuoka[3], Kenji Tsushima[4], Koichiro Tatsumi[1]

1 Department of Respirology, Graduate School of Medicine, Chiba University, Chiba-city, Chiba, Japan, 2 Department of Diagnostic Radiology and Radiation Oncology, Graduate School of Medicine, Chiba University Hospital, Chiba-city, Chiba, Japan, 3 Department of Radiology, St. Marianna University School of Medicine, Kawasaki-city, Kanagawa, Japan, 4 Department of Pulmonary Medicine, International University of Health and Welfare, School of Medicine, Kozunomori, Narita-city, Chiba, Japan

* suzuki-m.7447@chiba-u.jp

## Abstract

### Background

Combined pulmonary fibrosis with emphysema (CPFE) is a clinically meaningful syndrome characterized by coexisting upper-lobe emphysema and lower-lobe interstitial fibrosis. However, ambiguous diagnostic criteria and, particularly, the absence of objective methods to quantify emphysematous/fibrotic lesions in patients with CPFE confound the interpretation of the pathophysiology of this syndrome. We analyzed the relationship between objectively quantified computed tomography (CT) measurements and the results of pulmonary function testing (PFT) and clinical events in CPFE patients.

### Materials and methods

We enrolled 46 CPFE patients who underwent CT and PFT. The extent of emphysematous lesions was obtained by calculating the percent of low attenuation area (%LAA). The extent of fibrotic lesions was calculated as the percent of high attenuation area (%HAA). %LAA and %HAA values were combined to yield the percent of abnormal area (%AA). We assessed the relationships between CT parameters and other clinical indices, including PFT results. Multivariate analysis was performed to examine the association between the CT parameters and clinical events.

### Results

A greater negative correlation with percent predicted diffusing capacity of the lung for carbon monoxide ($DL_{CO}$ %predicted) existed for %AA ($r = -0.73$, $p < 0.001$) than for %LAA or %HAA alone. The %HAA value was inversely correlated with percent predicted forced vital

**Data Availability Statement:** All relevant data are within the paper and its Supporting Information files (S1 Appendix).

**Funding:** NK recieved the grants from the Ministry of Education, Science, Sports and Culture, Grant-in-Aid for Scientific Research (C) (16K01407,19K12816), the Chiba Foundation for Health Promotion & Disease Prevention(No.1272). Koichiro T recieved the grants from the Respiratory Failure Research Group (H26-Intractable diseases-General-076) from the Ministry of Health, Labour and Welfare, Japan. The funders had no role in study design, data collection and analysis, decision to publish, or preparation of the manuscript.

**Competing interests:** The authors have declared that no competing interests exist.

capacity (r = -0.48, p < 0.001), percent predicted total lung capacity (r = -0.48, p < 0.01), and $DL_{CO}$ %predicted (r = -0.47, p < 0.01). Multivariate logistic regression analysis found that % AA showed the strongest association with hospitalization events (odds ratio = 1.20, 95% confidence interval = 1.01–1.54, p = 0.029).

## Conclusion

Quantitative CT measurements reflected deterioration in pulmonary function and were associated with hospitalization in patients with CPFE. This approach could serve as a useful method to determine the extent of lung morphology, pathophysiology, and the clinical course of patients with CPFE.

## Background

Combined pulmonary fibrosis with emphysema (CPFE) is characterized by the coexistence of upper-lobe emphysema and lower-lobe fibrosis. Cottin et al. proposed the term "CPFE" for the entity characterized by these radiological features, a preserved lung volume, and severely diminished capacity for gas exchange [1]. The coexistence of emphysema and pulmonary fibrosis in individual patients is not rare in clinical practice. The estimated prevalence of pulmonary fibrosis is 8% in patients with stage 2 or higher chronic obstructive pulmonary disease (COPD) [2], and approximately 30% of patients with idiopathic pulmonary fibrosis (IPF) show manifestations of emphysema on chest computed tomography (CT) [3]. CPFE usually develops in men who are commonly older than 60 years of age and are current or ex-smokers, which is similar to the epidemiology of IPF [4, 5]. Follow-up studies have found that the prognosis of patients with CPFE is very poor, because of complications such as severe pulmonary hypertension, lung cancer, and acute exacerbations (AEs) [3, 6–8].

Patients with CPFE show severely reduced diffusing capacity of the lung for carbon monoxide ($DL_{CO}$), mild airflow limitation, and preserved lung capacity [9]. The baseline value for percent predicted $DL_{CO}$ ($DL_{CO}$ %predicted) is lower in patients with CPFE than in patients with COPD without pulmonary fibrosis [10] or in patients with IPF alone [11]. The respective annual decreases in vital capacity (VC) and $DL_{CO}$ were significantly smaller and larger in patients with CPFE than in patients with IPF [11]. The findings on pulmonary function testing (PFT) of CPFE patients are the results of additive effects or the counterbalance between the restrictive effects of pulmonary fibrosis and the hyperinflation associated with emphysematous lesions [9].

On the other hand, for patients with CPFE, several studies have reported on the correlations between radiological morphological findings and pulmonary function [12, 13]. Subjective visual assessments of emphysematous/fibrotic lesions on radiological imaging have been commonly performed. However, objective quantitative assessments are needed to clarify the relationships between morphological findings and the physiological characteristics of the disease, and to allow reproducible and longitudinal multicenter studies. A few recent studies used multidetector computed tomography (MDCT) to perform simultaneous objective quantitative assessments of the extent of emphysema and fibrosis in patients with CPFE and determined the association between the morphological findings and pulmonary dysfunction [14, 15]. However, whether these objective assessments are associated with the clinical manifestations of patients with CPFE has not been established.

Thus, the aims of this study were as follows: first, to evaluate the correlations between PFT results, including $DL_{CO}$, and the extent of emphysematous/fibrotic lesions as reflected by quantitative CT measurements; and second, to assess these CT measurements in relation to the factors associated with clinical events in patients with CPFE.

## Materials and methods

### Participants

The study protocol conformed to the Declaration of Helsinki and was approved by the Ethics Committee of our university (approval numbers: 857 and 2083), and written informed consent was obtained from all study participants.

We enrolled 113 patients who presented to our hospital between July 2012 and August 2018 for CPFE management. CPFE was diagnosed according to CT criteria described by Cottin et al., as follows: 1) the presence of a predominantly upper lobe emphysema, defined as well-demarcated areas of decreased attenuation, marginated by a very thin (< 1 mm) or no wall, and/or multiple bullae (> 1 cm); and 2) the presence of predominantly peripheral and basal pulmonary fibrosis, defined as reticular opacities and traction bronchiectasis with or without honeycombing [1]. Emphysema was classified into the following 3 groups: centrilobular, paraseptal, and mixed type (centrilobular plus paraseptal) emphysema. The patterns of pulmonary fibrosis were classified according to the IPF guidelines into usual interstitial pneumonia (UIP), probable UIP, indeterminate for UIP, and alternative-diagnosis pattern [5]. The classification of the patterns of pulmonary emphysema and fibrosis were performed independently by 3 clinical pulmonologists (MS, NK, and JI) who were blinded to the clinical information of the study patients.

We finally enrolled 46 patients with CPFE (Fig 1) after eliminating those with the following exclusion criteria: 1) lung cancer (n = 24); 2) connective tissue disease (n = 18); 3) systemic glucocorticoid treatment (n = 4); 4) infectious disease, including mycobacterial disease and aspergillosis (n = 3); 5) thoracic surgery (n = 2); 6) chemotherapy (n = 2); 7) heart failure (n = 0); and 8) others (n = 14).

### MDCT examinations

All patients underwent 64-MDCT (Aquilion ONE and Aquilion PRIME; Canon Medical Systems, Tokyo, Japan) and were scanned from the thoracic inlet to the diaphragm during full

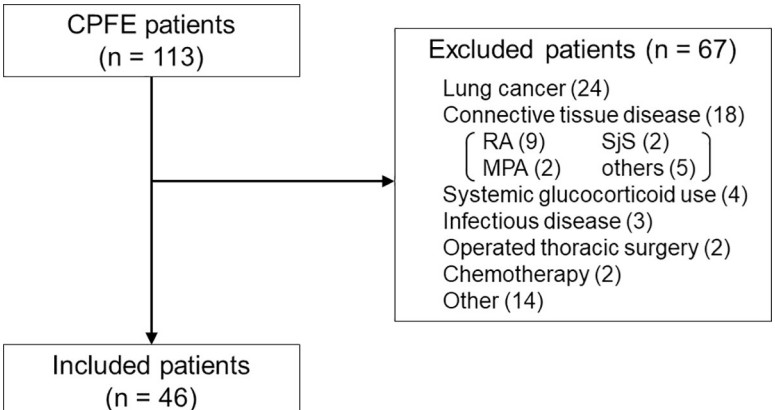

**Fig 1. Study population.** A total of 46 patients with combined pulmonary fibrosis with emphysema were included. Abbreviations: CPFE, combined pulmonary fibrosis with emphysema; RA, rheumatoid arthritis; SjS, Sjögren syndrome; MPA, microscopic polyangiitis.

inspiration without contrast enhancement. The MDCT scan parameters were as follows: collimination, 0.5 mm; 120 kV; auto-exposure control; gantry rotation time, 0.5 second; and beam pitch, 0.83. All images were reconstructed by standard algorithms (FC07) with a slice thickness of 0.5 mm and a reconstruction interval of 0.5 mm. The voxel size was $0.63 \times 0.63 \times 0.5$ mm.

## Quantitative CT measurements

We selected 4 CT slices from each CT series as follows: the first upper lung slice was taken 1 cm above the upper margin of the aortic arch (upper lesion), the second upper lung slice was taken at the carina (middle lesion), the first lower lung slice was taken 1 cm below the right inferior pulmonary vein (lower lesion), and the second lower lung slice was taken at the lower edge of the heart (bottom lesion). High-resolution computed tomography (HRCT) images were analyzed by an image-processing program (ImageJ, version 1.51j8, available at http://rsb.info.nih.gov/ij/; National Institutes of Health, Bethesda, MD, USA).

The quantitative measurements were performed in accordance with Matsuoka et al. [14, 16]. We used a threshold technique to segment all the pixels between -200 and -1024 Hounsfield units (HU) as the total lung area (TLA). To determine the extent of emphysema, we segmented pixels lower than -950 HU as the low attenuation area (LAA; Fig 2A and 2B), and %LAA was calculated for each slice as the percent of LAA relative to TLA. Similarly, to determine the extent of interstitial fibrotic lesions, we segmented all the pixels greater than -700 HU as the high attenuation area (HAA; Fig 2C and 2D); and %HAA was calculated for each slice as the percent of HAA relative to TLA. To calculate the total area of emphysematous and fibrotic changes, the percent of abnormal area (%AA) was obtained by summing the values for %LAA and %HAA at each thoracic level. The %LAA, %HAA, and %AA were calculated as the mean values of %LAA, %HAA, and %AA in each CT slice, respectively. These data were confirmed independently by 2 pulmonologists (MS and NK). All data were anonymized, and the observers were blinded to other characteristics of the participants when the imaging analysis was performed.

## Pulmonary function testing

All study participants underwent PFT by a CHSTAC-8900 spirometer (Chest MI, Tokyo, Japan). PFT was performed in accordance with the guidelines of the American Thoracic

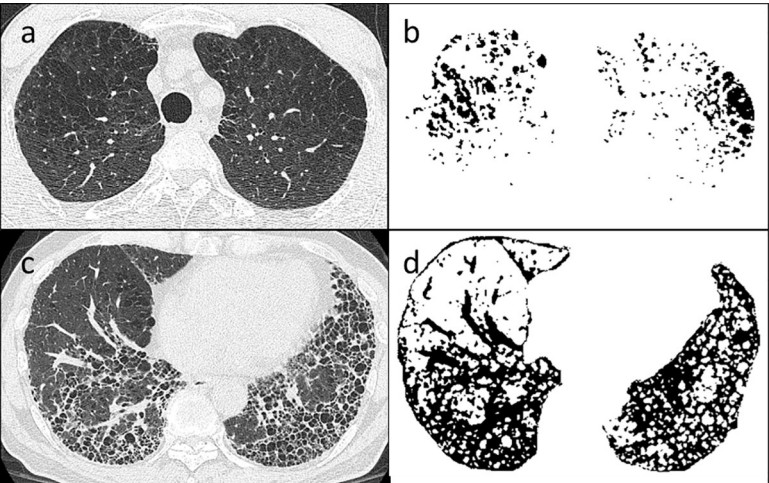

**Fig 2.** Axial computed tomography (CT) images of the upper (a) and lower lungs (c). Pixels with attenuation values between -950 and -1024 Hounsfield units (HUs) (b) and greater than -700 HUs (d) are highlighted in black on the CT scans.

Society and European Respiratory Society [17]. Total lung volume was determined by the helium dilution method, and $DL_{CO}$ and alveolar ventilation were determined by the single-breath method. The values for percent predicted forced expiratory volume in 1 second ($FEV_1$%predicted) were calculated according to the equations of the Japanese Respiratory Society [18].

### Clinical events

We investigated clinical events (hospitalizations, AEs, and deaths) during the observation period. The diagnosis of AEs was based on the AE criteria for IPF, as follows [19]: 1) acute worsening or development of dyspnea, typically < 1 month duration, 2) new bilateral ground-glass opacity and/or consolidation on CT, and 3) deterioration not fully explained by cardiac failure or fluid overload. In addition, duration of follow-up, smoking history, and serum levels of Krebs von den Lungen-6 (KL-6) were determined.

### Statistical analysis

Results are expressed as means ± standard deviation (± SD) or as medians (interquartile range [IQR]) as appropriate. Intraclass correlation coefficients (ICC) were determined to assess the intra- and interobserver reliability of CT measurements. After confirmation of the normality of the data on study parameters, the correlations between CT measurements and the results of PFT were assessed by Spearman rank correlation analysis, as appropriate. Comparisons between the group of patients with and without hospitalization events were performed by the Mann-Whitney *U* test. Multivariate regression analysis was performed to identify which variables were associated with hospitalization events. The variables that were significant in the univariate model were then entered into a multivariate regression analysis to identify the independent determinants of hospitalization events. For all analyses, the null hypothesis was rejected at the 5% level. Statistical analysis was performed by JMP 13.0 software (SAS Institute, Cary, NC).

## Results

### Patient characteristics

The general characteristics of the 46 enrolled CPFE patients are shown in Table 1. The mean age was 67.2 ± 7.8 years, and the 97.8% of patients were males. The mean body mass index (BMI) was 24.0 ± 3.7 $kg/m^2$. Except for 1 participant, all patients had a smoking history, and a mean of 62.0 ± 41.2 pack-years. The mean duration of follow-up was 1,087.9 ± 574.5 days.

### PFT and CT measurements

The results of PFT and quantitative CT measurements are presented in Table 1. The mean percent predicted forced vital capacity (FVC %predicted), the mean $FEV_1$%predicted, the mean percent predicted total lung capacity (TLC %predicted), and the mean $DL_{CO}$ %predicted were 86.7 ± 20.3%, 81.8 ± 18.3%, 84.3 ± 15.9%, and 65.4 ± 23.3%, respectively.

Study participants were classified by emphysema type as centrilobular (n = 21), paraseptal (n = 17), and mixed (n = 8) type. Regarding the pattern of pulmonary fibrosis, the patients were categorized as UIP (n = 15), probable UIP (n = 19), indeterminate for UIP pattern (n = 10), and alternative diagnosis (n = 2).

The median (IQR) %LAA, %HAA, and %AA were 4.9% (1.9–8.5%), 20.2% (15.4–25.6%), and 25.5% (21.0–36.2%), respectively. The CT measurements showed excellent reproducibility. The ICC for intraobserver variability among the CT parameters were as follows: %LAA, 0.998

**Table 1. Patient characteristics, pulmonary function tests, and computed tomography measurements.**

| | Mean ± SD |
|---|---|
| Age (years) | 67.2 ± 7.8 |
| Male/female, n (%) | 45 (97.8%)/1 (2.2%) |
| BMI (kg/m²) | 24.0 ± 3.7 |
| Pack years | 62.0 ± 41.2 |
| Smoker (Current or ever/never) | 45 (97.8%)/1 (2.2%) |
| Follow-up duration (days) | 1087.9 ± 574.5 |
| KL-6 (U/mL) | 1038.8 ± 1041.0 |
| Pulmonary function tests | |
| FVC (L) | 3.1 ± 0.8 |
| FVC %predicted (%) | 86.7 ± 20.3 |
| $FEV_1$%predicted (%) | 81.8 ± 18.3 |
| $FEV_1$/FVC (%) | 77.9 ± 9.3 |
| FRC %predicted (%) | 81.8 ± 16.6 |
| RV %predicted (%) | 83.3 ± 21.2 |
| TLC (L) | 4.7 ± 1.0 |
| TLC %predicted (%) | 84.3 ± 15.9 |
| $DL_{CO}$ (mL/min/mmHg) | 11.7 ± 4.6 |
| $DL_{CO}$ %predicted (%) | 65.4 ± 23.3 |
| CT measurements | |
| Emphysema type (centrilobular/paraseptal/mixed type) | 21/17/8 |
| Fibrosis type (UIP/probable UIP/indeterminate for UIP/alternative diagnosis pattern) | 15/19/10/2 |
| %LAA (median [IQR]) (%) | 4.9 (1.9–8.5) |
| %HAA (median [IQR]) (%) | 20.2 (15.4–25.6) |
| %AA (median [IQR]) (%) | 25.5 (21.0–36.2) |
| Clinical events | |
| Hospitalization, n (%) | 8 (17.4%) |
| Acute exacerbation, n (%) | 2 (4.3%) |
| Death, n (%) | 3 (6.5%) |

Abbreviations: BMI, body mass index; KL-6, Krebs von den Lungen-6; UIP, usual interstitial pneumonia; FVC, forced vital capacity; $FEV_1$, forced expiratory volume in 1 second; FRC, functional residual capacity; RV, residual volume; TLC, total lung capacity; $DL_{CO}$, diffusing capacity of the lung for carbon monoxide; %LAA, percent of low attenuation area to total lung area; %HAA, percent of high attenuation area to total lung area; %AA, percent of abnormal area to total lung area.

(95% confidence interval [CI], 0.997–0.999); %HAA, 0.998 (0.996–0.999); %AA, 0.998 (0.996–0.999). The ICC between 2 observers (interobserver variability) for the CT parameters were as follows: %LAA, 0.971 (95% CI, 0.948–0.984); %HAA, 0.994 (0.989–0.997); %AA, 0.984 (0.971–0.991).

The correlations between the CT measurements and the results of PFT are shown in Table 2. The %AA parameter was more negatively correlated with $DL_{CO}$ %predicted (r = -0.73, p < 0.001) than %LAA (r = -0.51, p < 0.001) or %HAA (r = -0.47, p < 0.01) alone. The %AA parameter was also significantly correlated with FVC %predicted, $FEV_1$%predicted, and TLC %predicted. The %HAA parameter was inversely correlated with FVC %predicted, $FEV_1$/FVC, TLC %predicted, and $DL_{CO}$ %predicted. In contrast, no significant correlations between %LAA and the results of PFT existed, except $DL_{CO}$ %predicted. Fig 3 demonstrates the correlations between $DL_{CO}$ %predicted and CT measurements on a two-dimensional analysis plot.

**Table 2. Correlations between computed tomography measurements and results of pulmonary function tests.**

| | %LAA | | %HAA | | %AA | |
|---|---|---|---|---|---|---|
| | r | p value | r | p value | r | p value |
| FVC %predicted | -0.06 | 0.69 | -0.48 | < 0.001 | -0.46 | < 0.01 |
| FEV$_1$%predicted | -0.29 | 0.05 | -0.29 | 0.05 | -0.45 | < 0.01 |
| FEV$_1$/FVC | -0.23 | 0.12 | 0.29 | < 0.05 | 0.13 | 0.41 |
| FRC %predicted | 0.11 | 0.49 | -0.41 | < 0.01 | -0.30 | 0.06 |
| RV %predicted | 0.2 | 0.22 | -0.5 | < 0.01 | -0.27 | 0.10 |
| TLC %predicted | -0.02 | 0.89 | -0.48 | < 0.01 | -0.46 | < 0.01 |
| DL$_{CO}$ %predicted | -0.51 | < 0.001 | -0.47 | < 0.01 | -0.73 | < 0.001 |

Abbreviations: %LAA, percent of low attenuation area to total lung area; %HAA, percent of high attenuation area to total lung area; %AA, percent of abnormal area to total lung area; FVC, forced vital capacity; FEV$_1$, forced expiratory volume in 1 second; FRC, functional residual capacity; RV, residual volume; TLC, total lung capacity; DL$_{CO}$, diffusing capacity of the lung for carbon monoxide.

## Clinical events

Table 1 also shows a summary of the clinical events. Three patients died of AE of IPF, chronic respiratory dysfunction, or heart failure. Two patients developed an AE. They received systemic corticosteroids and noninvasive oxygenation therapy; 1 patient died and the other recovered. Eight patients were hospitalized for the following reasons: bacterial pneumonia (n = 2), heart failure due to an old myocardial infarction (n = 2), AEs of IPF (n = 2), pneumothorax (n = 1), and other (n = 1). The difference between the duration of follow-up for the participants with or without hospitalization was not significant (median duration 1187 days [627–2373 days] vs 1082 days [21–2255 days]; p = 0.27) (S1 Appendix).

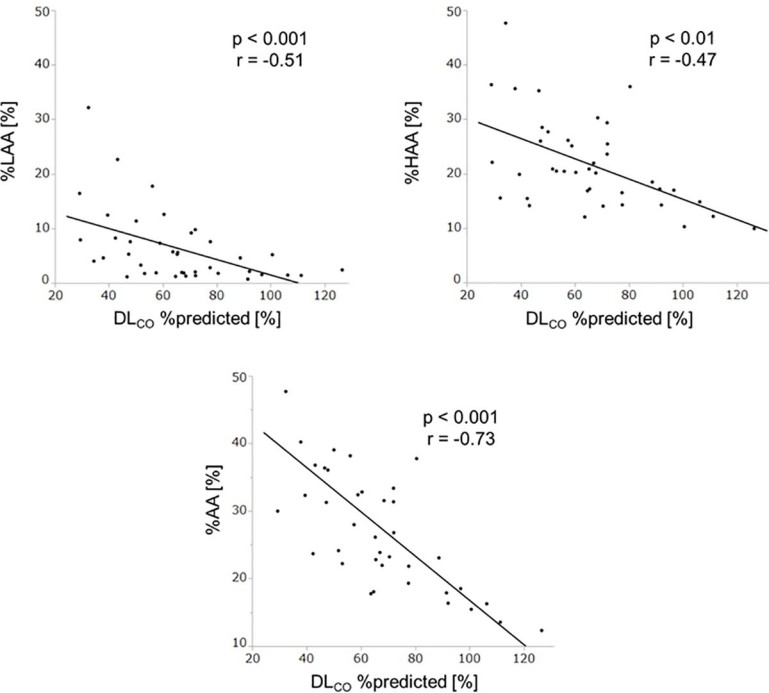

**Fig 3. Correlations of percent predicted diffusing capacity of the lung for carbon monoxide with the percent of low attenuation area, the percent of high attenuation area, and the percent of abnormal area.** Abbreviations: DL$_{CO}$, diffusing capacity of the lung for carbon monoxide; %LAA, percent of low attenuation area; %HAA, percent of high attenuation area; %AA, percent of abnormal area.

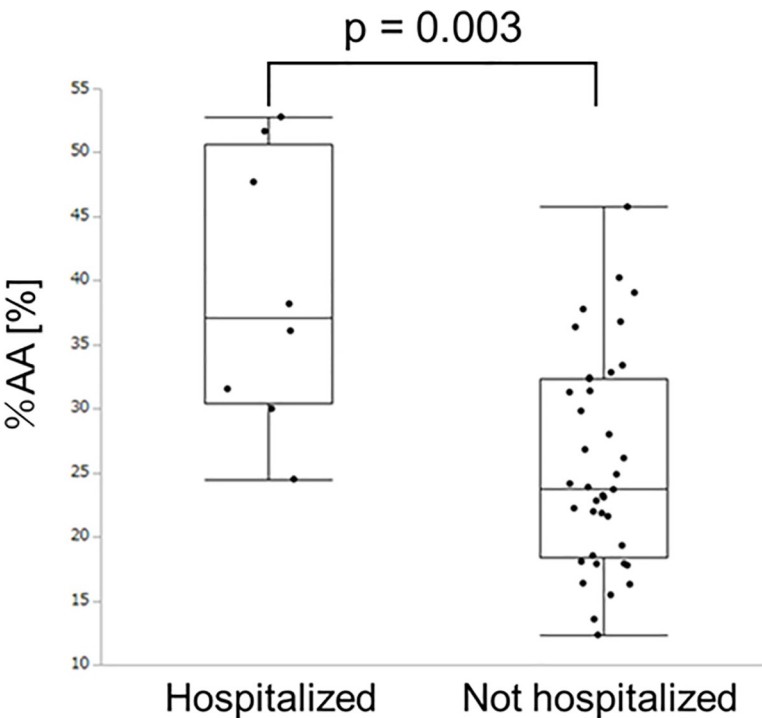

**Fig 4. Comparison of the percents of abnormal lung areas in participants with vs those without hospitalization.**
Abbreviation: %AA, percent of abnormal area.

## Clinical events and CT measurements

The hospitalization events were related to the parameter, %AA (Fig 4). The results of univariate and multivariate analysis for clinical events are shown in Table 3. The univariate analysis showed that pack-years (odds ratio [OR] = 0.97, 95% CI = 0.93–0.99, p = 0.035), $DL_{CO}$ %predicted (OR = 0.91, 95% CI = 0.82–0.96, p = 0.0006), and %AA (OR = 1.16, 95% CI = 1.06–1.32, p = 0.0005) were significantly associated with hospitalization. Multivariate analysis identified %AA as the only independent factor associated with hospitalization (OR = 1.20, 95% CI = 1.01–1.54, p = 0.029).

## Discussion

The key points of this study are as follows: first, the objective quantified CT values for emphysematous LAA and fibrotic HAA were significantly correlated with evidence of impaired

**Table 3. Univariate and multivariate analysis for hospitalization events.**

|  | Univariate | | Multivariate | |
|---|---|---|---|---|
|  | Odds ratio (95% CI) | p value | Odds ratio (95% CI) | p value |
| Age | 1.00 (0.91–1.11) | 0.95 | | |
| BMI | 0.91 (0.74–1.11) | 0.33 | | |
| Pack years | 0.97 (0.93–0.99) | 0.035 | 0.97 (0.90–1.01) | 0.1733 |
| FVC %predicted | 1.07 (0.41–2.99) | 0.89 | | |
| $DL_{CO}$ %predicted | 0.91 (0.82–0.96) | 0.0006 | 0.96 (0.86–1.04) | 0.3629 |
| %AA | 1.16 (1.06–1.32) | 0.0005 | 1.20 (1.01–1.54) | 0.0290 |

Abbreviations: CI, confidence interval; BMI, body mass index; FVC, forced vital capacity; $DL_{CO}$, diffusing capacity of the lung for carbon monoxide; %AA, percent of abnormal area to total lung area.

pulmonary function; second, the radiological parameters obtained by MDCT were associated with the clinical course of patients with CPFE.

Although previous studies have demonstrated an association between radiological assessments and pulmonary function in patients with CPFE [12, 13, 20], the objective quantitative methods used for determining the simultaneous extents of emphysema and pulmonary fibrosis have not been fully explored. However, recent studies on quantitative assessment defined the LAA associated with emphysematous lesions as the regions of lung density that are lower than the threshold of -950 HU, and the HAA associated with fibrotic lesions as the regions of lung density that are higher than the threshold of -700 HU [14–16]. Here, we used those CT criteria and confirmed that the CT parameters were significantly correlated with the results of PFT. These data suggest that this quantitative method may be reproducible and might enable us to perform an objective evaluation of abnormal CT findings.

Among the relationships between CT parameters and PFT results, the abnormal area (% AA), which was defined as the sum of %LAA and %HAA, had a greater correlation with a decrease in $DL_{CO}$ %predicted than %LAA or %HAA alone. The $DL_{CO}$ value is one of the most clinically valuable measurements based on the ability of the lungs to transfer gas from inhaled air to the red blood cells [21]. Although most PFT parameters of patients with CPFE have presented various patterns in accordance with specific morphological changes, a severely reduced $DL_{CO}$ value has been seen to be the most common abnormal finding [8, 9]. A decrease in the $DL_{CO}$ value of patients with CPFE is believed to be associated with alveolar destruction, loss of the pulmonary vascular bed, and alveolar wall thickening/collapse due to both emphysema and fibrotic changes [4]. Thus, the additive effect of emphysematous lesions and pulmonary fibrosis in the lungs of CPFE patients might account for the fact that %AA was more strongly associated with a decreased $DL_{CO}$ %predicted value than %LAA or %HAA alone. Additionally, in patients with IPF, decreased $DL_{CO}$ has been correlated with decreased exercise tolerance, including distance obtained during the 6-minute-walk test [22]. Our results indicate that chest CT assessments might reflect pulmonary physiological dysfunction in patients with CPFE as well as in those with IPF.

In addition, %LAA was only correlated with $DL_{CO}$ %predicted, and tended to be associated with the severity of the obstructive impairment. Although a number of studies of patients with COPD have reported on the relationship between LAA and obstructive impairment [23], only a few reports have described the relationship between LAA and pulmonary dysfunctions in CPFE patients [12, 14]. They reported a correlation between %LAA and decreased $FEV_1$%predicted and between %LAA in the upper lung slice and a decline in $DL_{CO}$ %predicted [12, 14]. In patients with CPFE, the increased traction caused by pulmonary fibrosis prevents collapse of the expiratory airway and expiratory airflow limitation, which are associated with emphysema. Therefore, these patients show a preserved $FEV_1$, FRC %predicted, and RV %predicted [4, 9]. Meanwhile, our study showed that %HAA was more strongly correlated with FRC %predicted and RV %predicted than %LAA. Thus, we consider that the extent of emphysema affects the pulmonary function of patients with CPFE less than their fibrotic lesions affect pulmonary function.

Emphysema subtypes are divided into three types; centrilobular, paraseptal, and panlobular [24]. Centrilobular emphysema is commonly complicated by COPD [25], and emphysema subtypes in COPD are related to pulmonary symptoms, the development of lung cancer, and worsening radiological findings [26, 27]. In contrast, CPFE patients are likely to exhibit paraseptal emphysema (30%-65%) [28, 29], which is consistent with the present study. CPFE with paraseptal emphysema lesions is associated with CTD and a poor prognosis [30–32]. Considering emphysema subtypes in CPFE might contribute to the putative clinical course.

Another notable result of this study concerns our investigation of the relationship between objective quantitative CT measurements and the clinical events of patients with CPFE. Only

the parameter %AA was significantly independently associated with hospitalization events on multivariate regression analysis of such clinical factors as age, BMI, pack-years, FVC %predicted, DL$_{CO}$ %predicted, and %AA. Respiratory-related hospitalizations have prognostic significance for patients with COPD and IPF [33–35] and are independently associated with decreased FVC in patients with IPF [34]. A previous report has described the relationship between the radiological assessments and clinical events in CPFE [13]. This study used a subjective assessment of CT images that was based on a fibrosis-weighted CT index, and demonstrated a correlation with the outcome in CPFE patients. In our study, we used a simple automated analysis of CT images, and found a significant relationship between %AA and hospitalization. Our results suggest that this radiological assessment might reflect disease severity and have prognostic value for patients with CPFE.

Interestingly, in our study population, the %AA parameter was more relevant to hospitalization than the DL$_{CO}$ %predicted value. Although reduced DL$_{CO}$ is commonly observed in patients with CPFE, the impact of DL$_{CO}$ on outcome has not been clarified. In patients with IPF, not only gender, age, physiological stage, and composite physical index, but also a decrease from baseline DL$_{CO}$ %predicted, are known to be significant predictors of mortality [36]. A decrease in DL$_{CO}$ is also a predictor of exercise intolerance in patients with COPD [37, 38]. Previous studies have reported on the association between radiological assessments and clinical manifestations in IPF and COPD. In IPF patients with stable or exacerbated disease, the extent of fibrotic changes such as reticulation, honeycombing, and traction bronchiectasis has been reported to be a significant predictor for the risk of exacerbation and mortality [39, 40]. Similarly, the extent of emphysematous lung predicts the risk of exacerbation and mortality in patients with COPD [41, 42]. For our CPFE patients, we evaluated the %AA parameter that reflects both emphysematous and fibrotic lesions and found that %AA was an independent predictor for hospitalization.

The reasons for the hospitalization of our study patients varied; almost all hospitalizations were for respiratory- and circulatory-associated conditions such as pneumonia, pneumothorax, AE, and heart failure. These complications were similar to those reported for COPD and IPF patients [43, 44]. Severe pulmonary dysfunction leads to respiratory complications such as pneumonia, pneumothorax, and AEs in both COPD and IPF patients [19, 45, 46]. In patients with COPD, endothelial dysfunction and remodeling of the pulmonary vascular bed are considered to be associated with hypoxia and systemic inflammation, which might contribute to the development of cardiovascular disease [43, 47]. Patients with IPF have an increased risk of vascular disease in comparison with the general population [48]. In CPFE, the unravelling mechanisms in common with COPD and IPF could lead to these complications.

We also found a mild association between smoking pack-years and hospitalization events. Smoke-induced oxidative damage induces regenerating precursor cells in both IPF and COPD, which might lead to abnormal tissue remodeling and functional impairment in these diseases [49]. A positive smoking history has been found to increase mortality in patients with IPF [50]. In patients with COPD, the cessation of smoking reduces the risk of exacerbation [51] and subsequent mortality, even in patients with severe disease [52]. Furthermore, continuous smoking affects the progression of disease in patients with CPFE more strongly than former smoking does [53]. Therefore, the cessation of smoking is considered to essential for patients with CPFE.

Physiological measurements of lung function have been conventionally performed to evaluate disease severity in patients with lung disease. However, computer-based HRCT image analysis has greatly improved, and has enabled us to assess the extent of lung disease and to quantify morphological changes [54, 55]. The recently introduced densitometric- and histogram-based analysis of CT images provides data on mean lung attenuation, skewness, and

kurtosis. These parameters have been reported to show associations with parameters of pulmonary function and disease progression in various lung diseases [56]. In addition, some objective CT measurements have been found to be useful for the long-term monitoring of patients with emphysematous and fibrotic lung disease [25, 57, 58]. Quantitative CT assessments, as well as other biomarkers, might be useful for the assessment of disease severity and prediction of the clinical course of patients with CPFE [59].

This study has limitations. First, this was a single-center study with a small number of enrolled patients and the pulmonary function of the subjects was preserved compared to previous studies [1, 4, 6, 12]. The numbers of AEs and hospitalizations were low compared with those numbers in previous studies of patients with COPD, IPF, or CPFE [7, 19, 42]. Second, samples were not taken from patients for a histopathological evaluation. Third, the inner spaces of honeycombing and airspace enlargement associated with fibrosis might have been assessed as LAA. However, reports on the measurement of honeycombing with a background of fibrotic lesions are rare, and methodologies for distinguishing between honeycombing and LAA have not been elucidated [56]. Additionally, pulmonary vessels were partially included in the HAA values. However, the indices of fibrosis and emphysema that were obtained using the quantitative methods in our study reflected pulmonary function and were associated with hospitalization events.

## Conclusion

In conclusion, objective quantitative CT measurements were significantly associated with the results of PFT and with the hospitalization of patients with CPFE. Quantitative CT measurements might serve as a useful method to determine the lung morphology, pathophysiology and clinical course of patients with CPFE.

## Supporting information

**S1 Appendix.**
(XLSX)

## Acknowledgments

We thank Dr. Akira Nishiyama, Mrs. Chieko Handa, Mrs. Tamie Hirano, and Mrs. Mika Sakurai for their technical assistance and general support.

## Author Contributions

**Conceptualization:** Masaki Suzuki, Naoko Kawata, Mitsuhiro Abe, Hajime Yokota, Jun Ikari, Shin Matsuoka, Kenji Tsushima, Koichiro Tatsumi.

**Data curation:** Masaki Suzuki, Naoko Kawata, Rie Anazawa, Yukiko Matsuura, Jun Ikari.

**Formal analysis:** Masaki Suzuki, Naoko Kawata, Hajime Yokota, Jun Ikari, Koichiro Tatsumi.

**Funding acquisition:** Naoko Kawata, Koichiro Tatsumi.

**Investigation:** Masaki Suzuki, Naoko Kawata, Hajime Yokota, Rie Anazawa, Yukiko Matsuura, Jun Ikari.

**Methodology:** Masaki Suzuki, Naoko Kawata, Mitsuhiro Abe, Hajime Yokota, Shin Matsuoka, Kenji Tsushima, Koichiro Tatsumi.

**Project administration:** Naoko Kawata, Hajime Yokota, Koichiro Tatsumi.

**Resources:** Naoko Kawata, Mitsuhiro Abe, Kenji Tsushima.

**Software:** Hajime Yokota.

**Supervision:** Naoko Kawata, Hajime Yokota, Kenji Tsushima.

**Validation:** Masaki Suzuki, Naoko Kawata, Shin Matsuoka.

**Visualization:** Masaki Suzuki, Hajime Yokota.

**Writing – original draft:** Masaki Suzuki.

**Writing – review & editing:** Masaki Suzuki, Naoko Kawata, Mitsuhiro Abe, Jun Ikari, Kenji Tsushima, Koichiro Tatsumi.

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
