## [Decision Letter · Decision Letter 0]

30 Jun 2020

PONE-D-20-02957

Objective quantitative multidetector computed tomography assessments in patients with combined pulmonary fibrosis with emphysema: relationship with pulmonary function and clinical events

PLOS ONE

Dear Dr. Suzuki,

Thank you for submitting your manuscript to PLOS ONE. After careful consideration, we feel that it has merit but does not fully meet PLOS ONE’s publication criteria as it currently stands. Therefore, we invite you to submit a revised version of the manuscript that addresses the points raised during the review process.

Please reply to the reviewers' comments and provide a point to point response.

We look forward to receiving your revised manuscript.

Kind regards,

Y. Peter Di, Ph.D.

Academic Editor

PLOS ONE

Journal Requirements:

Reviewers' comments:

Reviewer's Responses to Questions

**Comments to the Author**

1. Is the manuscript technically sound, and do the data support the conclusions?

Reviewer #1: Partly

Reviewer #2: Yes

Reviewer #3: Yes

2. Has the statistical analysis been performed appropriately and rigorously? 

Reviewer #1: Yes

Reviewer #2: Yes

Reviewer #3: Yes

3. Have the authors made all data underlying the findings in their manuscript fully available?

Reviewer #1: Yes

Reviewer #2: Yes

Reviewer #3: Yes

4. Is the manuscript presented in an intelligible fashion and written in standard English?

Reviewer #1: Yes

Reviewer #2: Yes

Reviewer #3: Yes

5. Review Comments to the Author

Reviewer #1: The study submitted by Masaki Suzuki et al. aims to evaluate the correlation between objective measurements of high-resolution chest CT and impaired pulmonary function, as well as clinical events in a prospective cohort of combined pulmonary fibrosis with emphysema (CPFE) patients. They found that the total abnormal area(%AA), %LAA and %HAA all correlated with decreased DLCO%predicted, %AA also showed a positive correlation with hospitalization, may have prognostic value.

In principle, the results are of interest for scientific and clinical community, deserved to be published, however I have some major concerns that prevent me to endorse its acceptance at the present stage.

1. The overall DLCO %predicted is 65.4 ± 23.3%, %LAA (median [IQR]) is only 4.9 (1.9 – 8.5) %. These two parameters reflect mild lung function impairment in this cohort. It is counterintuitive and does not go along with published data[1, 2], in which the DLCOs were usually severely reduced. I was wondering what is the participant recruitment strategy? (e.g., this research study only, patients with known pulmonary fibrosis and smoking history went on CT scanning, et al.) What is the source of potential biased selection?

2. Cardiac diseases especially heart failure should be added as exclusion criteria since lung edema would contribute to HAA value, add noise. Authors may carefully check if any patients with cardiac diseases had been enrolled in this study since 2 cases had been reported for hospitalization due to heart failure. It may be helpful to specify whether it is right-sided heart failure caused by chronic lung diseases.

3. Authors have defined -950 Hu as the threshold for low attenuation. It is adapted from Dr. Matsuoka’s study. However, 2mm slice thickness was used originally. Here using 0.5mm might have been too thin to measure LAA with a threshold setting of -950 Hu. It would overestimate emphysema[3]. Again, the %LAA in the entire cohort is around 5% with overestimation. The degree of emphysema makes negligible contribution to the lung function change. Therefore, it’s not surprising that no correlation found between %LAA with reduced LFT parameters and prognosis. Some of conclusions may due to bias.

1. Matsuoka, S., et al., Quantitative CT evaluation in patients with combined pulmonary fibrosis and emphysema: correlation with pulmonary function. Acad Radiol, 2015. 22(5): p. 626-31.

2. Ando, K., et al., Relationship between quantitative CT metrics and pulmonary function in combined pulmonary fibrosis and emphysema. Lung, 2013. 191(6): p. 585-91.

3. Gierada, D.S., et al., Effects of CT section thickness and reconstruction kernel on emphysema quantification relationship to the magnitude of the CT emphysema index. Acad Radiol, 2010. 17(2): p. 146-56.

Reviewer #2: I congratulate the authors for the research titled 'Objective quantitative multidetector computed tomography assessments in patients with combined pulmonary fibrosis with emphysema: relationship with pulmonary function and clinical event'. In the present study, CT findings and PFT values were compared in patients with CPFE, and the relationship between CT pathology (fibrozis and emphysema) and respiratory function loss was demonstrated. The research is very well organized and contributes to science in this regard. I think it can be published as it is.

Reviewer #3: 6/29/20

Editorial Office

PLOS | plos.org

1160 Battery Street, Suite 225, San Francisco, CA 94111

Re: Manuscript Number: PONE-D-20-02957

Objective quantitative multidetector computed tomography assessments in patients with combined pulmonary fibrosis with emphysema: relationship with pulmonary function and clinical events

Dear Editorial team:

The authors present a manuscript entitled “Objective quantitative multidetector computed tomography assessments in patients with combined pulmonary fibrosis with emphysema: relationship with pulmonary function and clinical events.” The authors should be commended on their efforts to utilize CT attenuation values in patients with combined pulmonary fibrosis with emphysema (CPFE), and to correlate these values with pulmonary function test parameters (PFTs) and clinical outcomes. The authors demonstrate that in CPFE, increased total areas of low and high attenuation tissue (total abnormal area) did correlate with worse PFTs parameters and worse clinical outcomes. On the one hand, it is not surprising that patients with greater abnormalities on chest CT have greater impairment in PFTs and worse clinical outcomes, and potentially, the authors merely confirmed this commonly held belief. On the other hand, their approach to correlate CT attenuation values in CPFE patients with PFTs and clinical outcomes is relevant to patients if this technological approach is widely available in the clinical setting, and the authors also nicely present data assessing the specific contribution of low versus high attenuation areas on PFT parameters. The study is limited by the relatively small number of patients and the small number of patients developing adverse outcomes, but the authors do address these limitations well.

Minor concerns:

1) In the Methods and in Figure 1, the authors exclude 18 patients with connective tissue disease (CTD). I do understand that CTD likely has a different mechanism of disease than patients with idiopathic interstitial pneumonia (IIP), but it seems to me that assessing attenuation values in combined emphysema and fibrosis in patients with CTD would not be much different than assessing attenuation values in patients with IIP. If CTD patients were included, the number of patients studied would obviously be higher. Would be helpful if the authors could address their thoughts on the reasons for excluding these patients from analysis in the study.

2) It would be helpful if the authors would include the absolute numerical values for the PFT parameters in Table 1 (ie, measured values in liters [L] for spirometry and lung volumes, and measured values in ml CO/min/mm Hg for DLCO). Percent predicted values by definition depend upon prediction formulas and thus predicted values, which can be widely variable. It is interesting that the DLCO percent predicted values overall were not as low as I would have expected, and this may be merely related to the predicted values which were utilized in this study.

3) Mechanisms leading to centrilobular emphysema versus paraseptal emphysema in patients with CPFE may be different. Centrilobular emphysema is almost certainly a consequence of cigarette smoking, but paraseptal emphysema may occur merely as a manifestation of extensive intra-parenchymal fibrosis. Thus, centrilobular versus paraseptal emphysema may have differing effects on PFTs and clinical outcomes. I doubt this can be further assessed in this study given the relatively small number of patients in total overall, but perhaps the authors could include in their discussion regarding the differing etiologies of centrilobular versus paraseptal emphysema in patients with pulmonary fibrosis.

6. PLOS authors have the option to publish the peer review history of their article (what does this mean?). If published, this will include your full peer review and any attached files.

Reviewer #1: No

Reviewer #2: No

Reviewer #3: No

---

## [Author Response · Author response to Decision Letter 0]

19 Aug 2020

Reviewer #1

Comment 1

1. The overall DLCO %predicted is 65.4 ± 23.3%, %LAA (median [IQR]) is only 4.9 (1.9 – 8.5) %. These two parameters reflect mild lung function impairment in this cohort. It is counterintuitive and does not go along with published data[1, 2], in which the DLCOs were usually severely reduced. I was wondering what is the participant recruitment strategy? (e.g., this research study only, patients with known pulmonary fibrosis and smoking history went on CT scanning, et al.) What is the source of potential biased selection?

Response

- We appreciate your pointing this out. As you mentioned, the patients with CPFE are reported to show a significant reduction of DLCO, and the values for DLCO in our subjects were relatively preserved compared to previous studies [1, 2]. Our study was conducted in a single center. The study population was enrolled in accordance with the criteria of CPFE proposed by Cottin et al., which has pulmonary fibrosis and interstitial fibrosis on CT [1]. In the present study, we carefully checked smoking history and physical examination, as well as CT findings by 3 pulmonologists with 10, 15, and 20 years of experience. Similarly, Matsuoka et al. and Ando et al. used the same criteria of CPFE used by Cottin et al. Matsuoka et al. analyzed the CPFE patients with %LAA > 5% and Ando et al. investigated the study population after screening. Additionally, in the previous study, subclinical CPFE categorized by radiologists and biopsy-proven CPFE exhibited preserved pulmonary function, including DLCO [3]. In contrast, approximately 13% of patients with IPF with emphysema had emphysema lesions > 5% [4]. This study was a single-center study with a small number of enrolled patients, which may be a potential bias. We took the average of four CT slices to reflect the impact of fibrosis on pulmonary function in the inferior aspects of the lungs. The progression of abnormal lesions on chest CT might depend on the selected CT indices.

In recent reports, IPF subjects with preserved %VC received greater benefit from antifibrotic agents [5], and nintedanib reduced the decline in pulmonary function and acute exacerbation risk not only in patients with IPF, but also in patients with chronic fibrosing ILDs [6, 7]. These data provide the possibility for CPFE patients with mild-to-moderate pulmonary dysfunction to respond to novel treatments. To confirm the verification of our study without a lower limit of %LAA, a multicentric cohort study is warranted.

We added study limitations as below.

In the Discussion section, lines 359–361:

“First, this was a single-center study with a small number of enrolled patients and the pulmonary function of the subjects was preserved compared to previous studies [1, 4, 6, 12].”

Comment 2

2. Cardiac diseases especially heart failure should be added as exclusion criteria since lung edema would contribute to HAA value, add noise. Authors may carefully check if any patients with cardiac diseases had been enrolled in this study since 2 cases had been reported for hospitalization due to heart failure. It may be helpful to specify whether it is right-sided heart failure caused by chronic lung diseases.

Response 2

- We appreciate your comment. As the reviewer pointed out, cardiac diseases, particularly cognitive heart failure, may affect values for HAA. We reviewed the cardiac status backgrounds in all our subjects at the time of enrollment. No subjects developed heart failure or pneumonia at the time of enrollment. In addition, no CT slices showed abnormal findings associated with pulmonary edema, such as ground glass opacity, bronchovascular bundle thickening with a central distribution and sparing of the lung cortex, and pleural effusion. In this study, two subjects developed heart failure due to old myocardial infarctions during the follow-up period. Unfortunately, because the two subjects were not examined with pulmonary artery catherization at the time of hospitalization, we could not conclude whether right- or left-sided heart failure was associated with chronic lung disease.

We revised the manuscript as below.

In the Materials and methods section, lines 108–109:

“6) chemotherapy (n = 2); 7) heart failure (n = 0); and 8) others (n = 14).”

In the Results section, lines 237–238:

“bacterial pneumonia (n = 2), heart failure due to an old myocardial infarction (n = 2),”

Comment 3

3. Authors have defined -950 Hu as the threshold for low attenuation. It is adapted from Dr. Matsuoka’s study. However, 2mm slice thickness was used originally. Here using 0.5mm might have been too thin to measure LAA with a threshold setting of -950 Hu. It would overestimate emphysema [3]. Again, the %LAA in the entire cohort is around 5% with overestimation. The degree of emphysema makes negligible contribution to the lung function change. Therefore, it’s not surprising that no correlation found between %LAA with reduced LFT parameters and prognosis. Some of conclusions may due to bias.

Response 3

- We appreciate your comments regarding CT slice thickness. We adopted a 0.5 mm thickness on CT for morphological evaluation in our previous reports [8–10]. Although CT images with various thicknesses (1–2.5 mm) were used for quantitative CT assessment [11–13], we adopted the same organized slice thickness for all subjects. We evaluated the correlation between the values obtained with 0.5 and 2 mm CT slice thicknesses. We prepared 2-mm thickness data stacked for the study population using ImageJ (https://imagej.net/Fiji). The %LAA and %HAA from the 2-mm CT thickness were calculated using the same method. The median (IQR) %LAA and %HAA from the 0.5- and 2-mm CT thicknesses were as follows: %LAA, 4.9 % (1.9–8.5) versus 4.7 % (1.7–8.4); and %HAA, 20.2 % (15.4–25.6) versus 19.4 % (15.4–25.6). The ICC of %LAA and %HAA between the 0.5- and 2-mm CT slice thicknesses were as follows: %LAA, 0.9993 (95% CI, 0.9987–0.9996); and %HAA, 0.9969 (0.9946–0.9983). The %LAA from the 0.5-mm thickness was larger than that from the 2-mm thickness. The coefficient values for both %LAA and %HAA were high. As pointed out, the slice thickness and the reconstruction kernel are quite important to perform quantitative measurements for imaging analysis. We plan to elucidate this point in a corollary study.

Reviewer #2

Comment

I congratulate the authors for the research titled 'Objective quantitative multidetector computed tomography assessments in patients with combined pulmonary fibrosis with emphysema: relationship with pulmonary function and clinical event'. In the present study, CT findings and PFT values were compared in patients with CPFE, and the relationship between CT pathology (fibrozis and emphysema) and respiratory function loss was demonstrated. The research is very well organized and contributes to science in this regard. I think it can be published as it is.

Response

- We greatly appreciate your helpful and encouraging comments. The pathogenesis and physiological findings are distinct in patients with CPFE, thus targeted antifibrotic therapy might be needed. We consider that larger clinical trials on CPFE are required to elucidate the relationship between pulmonary function, radiological characteristics, and clinical course.

Reviewer #3

Comment 1

1) In the Methods and in Figure 1, the authors exclude 18 patients with connective tissue disease (CTD). I do understand that CTD likely has a different mechanism of disease than patients with idiopathic interstitial pneumonia (IIP), but it seems to me that assessing attenuation values in combined emphysema and fibrosis in patients with CTD would not be much different than assessing attenuation values in patients with IIP. If CTD patients were included, the number of patients studied would obviously be higher. Would be helpful if the authors could address their thoughts on the reasons for excluding these patients from analysis in the study.

Response 1

- We appreciate your comments regarding CPFE with CTD. Compared to CPFE patients without CTD, CPFE patients with CTD were more often female gender, had deteriorated pulmonary function, pulmonary hypertension, and a poor prognosis [14, 15]. The mechanism of underlying CPFE with CTD might differ from that of patients with CPFE but without CTD and might be of clinical interest in clinical practice. Regarding clinical events, CPFE patients with CTD could have exacerbations and require hospitalization due to extra-pulmonary complications and treatment (immunosuppressants and/or steroids)-related side effects or infections. Thus, in focus on the relationship between CT assessments and clinical events of patients with CPFE but without CTD, we excluded CPFE patients with CTD in this study. We plan to investigate the differences between CPFE patients with or without CTD in a corollary study.

Comment 2

2) It would be helpful if the authors would include the absolute numerical values for the PFT parameters in Table 1 (ie, measured values in liters [L] for spirometry and lung volumes, and measured values in ml CO/min/mm Hg for DLCO). Percent predicted values by definition depend upon prediction formulas and thus predicted values, which can be widely variable. It is interesting that the DLCO percent predicted values overall were not as low as I would have expected, and this may be merely related to the predicted values which were utilized in this study.

Response 2

- We appreciate your comment. We revised Table 1 as below, including absolute numerical values for FVC, TLC, and DLCO. The absolute numerical values among our subjects were also lower than previous studies [1, 2, 4]; however, we enrolled the study population in accordance with the criteria of CPFE, as proposed by Cottin et al. [1]. Similarly, Matsuoka et al. and Ando et al. used the same criteria for CPFE. Matsuoka et al. analyzed CPFE patients with a %LAA > 5% and Ando et al. investigated the study population after screening. In contrast, approximately 13% of patients with IPF and emphysema had emphysema lesions > 5% [4]. The inclusion criteria might affect the pulmonary function differences in the study population. We plan to investigate the relationships between CT assessments and clinical courses in a multicentric cohort, including progressed CPFE subjects.

We revised Table 1 as below and the Supplemental file.

In the Results section, lines 188–189:

Table 1. Patient characteristics, pulmonary function tests, and computed tomography measurements

　 Mean ± SD

Age (years) 67.2 ± 7.8

Male/female, n (%) 45 (97.8 %)/1 (2.2 %)

BMI (kg/m2) 24.0 ± 3.7

Pack years 62.0 ± 41.2

Smoker (Current or ever/never) 45 (97.8 %)/1 (2.2 %)

Follow-up duration (days) 1087.9 ± 574.5

KL-6 (U/mL) 1038.8 ± 1041.0

Pulmonary function tests 

FVC (L) 3.1 ± 0.8

FVC %predicted (%) 86.7 ± 20.3

FEV1 %predicted (%) 81.8 ± 18.3

FEV1/FVC (%) 77.9 ± 9.3

FRC %predicted (%) 81.8 ± 16.6

RV %predicted (%) 83.3 ± 21.2

TLC (L) 4.7 ± 1.0

TLC %predicted (%) 84.3 ± 15.9

DLCO (mL/min/mmHg) 11.7 ± 4.6

DLCO %predicted (%) 65.4 ± 23.3

CT measurements 

Emphysema type (centrilobular/paraseptal/mixed type) 21/17/8

Fibrosis type (UIP/probable UIP/indeterminate for UIP/alternative diagnosis pattern) 15/19/10/2

%LAA (median [IQR]) (%) 4.9 (1.9 – 8.5)

%HAA (median [IQR]) (%) 20.2 (15.4 – 25.6)

%AA (median [IQR]) (%) 25.5 (21.0 – 36.2)

Clinical events 

Hospitalization, n (%) 8 (17.4 %)

Acute exacerbation, n (%) 2 (4.3 %)

Death, n (%) 3 (6.5 %)

Abbreviations: BMI, body mass index; KL-6, Krebs von den Lungen-6; UIP, usual interstitial pneumonia; FVC, forced vital capacity; FEV1, forced expiratory volume in 1 second; FRC, functional residual capacity; RV, residual volume; TLC, total lung capacity; DLCO, diffusing capacity of the lung for carbon monoxide; %LAA, percent of low attenuation area to total lung area; %HAA, percent of high attenuation area to total lung area; %AA, percent of abnormal area to total lung area.

Comment 3

3) Mechanisms leading to centrilobular emphysema versus paraseptal emphysema in patients with CPFE may be different. Centrilobular emphysema is almost certainly a consequence of cigarette smoking, but paraseptal emphysema may occur merely as a manifestation of extensive intra-parenchymal fibrosis. Thus, centrilobular versus paraseptal emphysema may have differing effects on PFTs and clinical outcomes. I doubt this can be further assessed in this study given the relatively small number of patients in total overall, but perhaps the authors could include in their discussion regarding the differing etiologies of centrilobular versus paraseptal emphysema in patients with pulmonary fibrosis.

Response 3

- We appreciate that you have pointed this out. Conventionally, emphysema is categorized into three types: centrilobular, paraseptal, and panlobular [16]. Centrilobular emphysema is characterized by low attenuation surrounded by normal attenuation located within the central portion of the pulmonary lobules. In contrast, paraseptal emphysema is characterized by low attenuation adjacent to subpleural and peribronchovascular areas. Centrilobular emphysema is commonly complicated by smoking, which is consistent with the finding that COPD patients often have centrilobular emphysema [17]. In addition, the emphysema subtypes are reportedly related to pulmonary symptoms, the development of lung cancer, and worsening radiological findings in patients with COPD [18, 19]. In contrast, CPFE patients frequently (30%-65%) exhibit paraseptal emphysema [2, 20], and CPFE with paraseptal emphysema is reportedly associated with CTD and a poor prognosis [14, 15, 21]. Similarly, our study population included 17 subjects (34%) with paraseptal emphysema, which is consistent with previous studies [2, 20]. Because our sample size was small, we could not perform a sub-group analysis in this study. We plan to investigate the relationship between emphysema types in combination with quantitative HRCT assessments in CPFE patients.

We revised the manuscript as below.

In the Discussion section, lines 300–306:

“Emphysema subtypes are divided into three types; centrilobular, paraseptal, and panlobular [24]. Centrilobular emphysema is commonly complicated by COPD [25], and emphysema subtypes in COPD are related to pulmonary symptoms, the development of lung cancer, and worsening radiological findings [26, 27]. In contrast, CPFE patients are likely to exhibit paraseptal emphysema (30%-65%) [28, 29], which is consistent with the present study. CPFE with paraseptal emphysema lesions is associated with CTD and a poor prognosis [30-32]. Considering emphysema subtypes in CPFE might contribute to the putative clinical course.”

References

1. Cottin V, Nunes H, Brillet PY, Delaval P, Devouassoux G, Tillie-Leblond I, et al. Combined pulmonary fibrosis and emphysema: a distinct underrecognised entity. The European respiratory journal. 2005;26(4):586-93. Epub 2005/10/06. doi: 10.1183/09031936.05.00021005. PubMed PMID: 16204587.

2. Kitaguchi Y, Fujimoto K, Hanaoka M, Kawakami S, Honda T, Kubo K. Clinical characteristics of combined pulmonary fibrosis and emphysema. Respirology (Carlton, Vic). 2010;15(2):265-71. Epub 2010/01/07. doi: 10.1111/j.1440-1843.2009.01676.x. PubMed PMID: 20051048.

3. Choi SH, Lee HY, Lee KS, Chung MP, Kwon OJ, Han J, et al. The value of CT for disease detection and prognosis determination in combined pulmonary fibrosis and emphysema (CPFE). PloS one. 2014;9(9):e107476. Epub 2014/09/10. doi: 10.1371/journal.pone.0107476. PubMed PMID: 25203455; PubMed Central PMCID: PMCPMC4159339.

4. Ryerson CJ, Hartman T, Elicker BM, Ley B, Lee JS, Abbritti M, et al. Clinical features and outcomes in combined pulmonary fibrosis and emphysema in idiopathic pulmonary fibrosis. Chest. 2013;144(1):234-40. Epub 2013/02/02. doi: 10.1378/chest.12-2403. PubMed PMID: 23370641.

5. Azuma A, Taguchi Y, Ogura T, Ebina M, Taniguchi H, Kondoh Y, et al. Exploratory analysis of a phase III trial of pirfenidone identifies a subpopulation of patients with idiopathic pulmonary fibrosis as benefiting from treatment. Respiratory research. 2011;12(1):143. Epub 2011/11/01. doi: 10.1186/1465-9921-12-143. PubMed PMID: 22035508; PubMed Central PMCID: PMCPMC3216874.

6. Flaherty KR, Wells AU, Cottin V, Devaraj A, Walsh SLF, Inoue Y, et al. Nintedanib in Progressive Fibrosing Interstitial Lung Diseases. The New England journal of medicine. 2019;381(18):1718-27. Epub 2019/10/01. doi: 10.1056/NEJMoa1908681. PubMed PMID: 31566307.

7. Cottin V, Wollin L, Fischer A, Quaresma M, Stowasser S, Harari S. Fibrosing interstitial lung diseases: knowns and unknowns. European respiratory review : an official journal of the European Respiratory Society. 2019;28(151). Epub 2019/03/01. doi: 10.1183/16000617.0100-2018. PubMed PMID: 30814139.

8. Anazawa R, Kawata N, Matsuura Y, Ikari J, Tada Y, Suzuki M, et al. Longitudinal changes in structural lung abnormalities using MDCT in chronic obstructive pulmonary disease with asthma-like features. PloS one. 2019;14(12):e0227141. Epub 2019/12/31. doi: 10.1371/journal.pone.0227141. PubMed PMID: 31887184; PubMed Central PMCID: PMCPMC6936827.

9. Takayanagi S, Kawata N, Tada Y, Ikari J, Matsuura Y, Matsuoka S, et al. Longitudinal changes in structural abnormalities using MDCT in COPD: do the CT measurements of airway wall thickness and small pulmonary vessels change in parallel with emphysematous progression? International journal of chronic obstructive pulmonary disease. 2017;12:551-60. Epub 2017/03/01. doi: 10.2147/copd.s121405. PubMed PMID: 28243075; PubMed Central PMCID: PMCPMC5315203.

10. Suzuki T, Tada Y, Kawata N, Matsuura Y, Ikari J, Kasahara Y, et al. Clinical, physiological, and radiological features of asthma-chronic obstructive pulmonary disease overlap syndrome. International journal of chronic obstructive pulmonary disease. 2015;10:947-54. Epub 2015/06/02. doi: 10.2147/copd.s80022. PubMed PMID: 26028967; PubMed Central PMCID: PMCPMC4440433.

11. Matsuoka S, Washko GR, Dransfield MT, Yamashiro T, San Jose Estepar R, Diaz A, et al. Quantitative CT measurement of cross-sectional area of small pulmonary vessel in COPD: correlations with emphysema and airflow limitation. Academic radiology. 2010;17(1):93-9. Epub 2009/10/03. doi: 10.1016/j.acra.2009.07.022. PubMed PMID: 19796970; PubMed Central PMCID: PMCPMC2790546.

12. Matsuoka S, Washko GR, Yamashiro T, Estepar RS, Diaz A, Silverman EK, et al. Pulmonary hypertension and computed tomography measurement of small pulmonary vessels in severe emphysema. American journal of respiratory and critical care medicine. 2010;181(3):218-25. Epub 2009/10/31. doi: 10.1164/rccm.200908-1189OC. PubMed PMID: 19875683; PubMed Central PMCID: PMCPMC2817812.

13. Matsuoka S, Yamashiro T, Diaz A, Estépar RS, Ross JC, Silverman EK, et al. The relationship between small pulmonary vascular alteration and aortic atherosclerosis in chronic obstructive pulmonary disease: quantitative CT analysis. Academic radiology. 2011;18(1):40-6. Epub 2010/10/16. doi: 10.1016/j.acra.2010.08.013. PubMed PMID: 20947389; PubMed Central PMCID: PMCPMC3006041.

14. Champtiaux N, Cottin V, Chassagnon G, Chaigne B, Valeyre D, Nunes H, et al. Combined pulmonary fibrosis and emphysema in systemic sclerosis: A syndrome associated with heavy morbidity and mortality. Seminars in arthritis and rheumatism. 2019;49(1):98-104. Epub 2018/11/10. doi: 10.1016/j.semarthrit.2018.10.011. PubMed PMID: 30409416.

15. Cottin V, Nunes H, Mouthon L, Gamondes D, Lazor R, Hachulla E, et al. Combined pulmonary fibrosis and emphysema syndrome in connective tissue disease. Arthritis and rheumatism. 2011;63(1):295-304. Epub 2010/10/12. doi: 10.1002/art.30077. PubMed PMID: 20936629.

16. Lynch DA, Austin JH, Hogg JC, Grenier PA, Kauczor HU, Bankier AA, et al. CT-Definable Subtypes of Chronic Obstructive Pulmonary Disease: A Statement of the Fleischner Society. Radiology. 2015;277(1):192-205. Epub 2015/05/12. doi: 10.1148/radiol.2015141579. PubMed PMID: 25961632; PubMed Central PMCID: PMCPMC4613878.

17. Smith BM, Austin JH, Newell JD, Jr., D'Souza BM, Rozenshtein A, Hoffman EA, et al. Pulmonary emphysema subtypes on computed tomography: the MESA COPD study. The American journal of medicine. 2014;127(1):94.e7-23. Epub 2014/01/05. doi: 10.1016/j.amjmed.2013.09.020. PubMed PMID: 24384106; PubMed Central PMCID: PMCPMC3882898.

18. Park J, Hobbs BD, Crapo JD, Make BJ, Regan EA, Humphries S, et al. Subtyping COPD by Using Visual and Quantitative CT Imaging Features. Chest. 2020;157(1):47-60. Epub 2019/07/10. doi: 10.1016/j.chest.2019.06.015. PubMed PMID: 31283919; PubMed Central PMCID: PMCPMC6965698.

19. Mouronte-Roibás C, Fernández-Villar A, Ruano-Raviña A, Ramos-Hernández C, Tilve-Gómez A, Rodríguez-Fernández P, et al. Influence of the type of emphysema in the relationship between COPD and lung cancer. International journal of chronic obstructive pulmonary disease. 2018;13:3563-70. Epub 2018/11/23. doi: 10.2147/copd.s178109. PubMed PMID: 30464438; PubMed Central PMCID: PMCPMC6214583.

20. Ciccarese F, Attinà D, Zompatori M. Combined pulmonary fibrosis and emphysema (CPFE): what radiologist should know. La Radiologia medica. 2016;121(7):564-72. Epub 2016/02/20. doi: 10.1007/s11547-016-0627-4. PubMed PMID: 26892068.

21. Todd NW, Jeudy J, Lavania S, Franks TJ, Galvin JR, Deepak J, et al. Centrilobular emphysema combined with pulmonary fibrosis results in improved survival. Fibrogenesis & tissue repair. 2011;4(1):6. Epub 2011/02/18. doi: 10.1186/1755-1536-4-6. PubMed PMID: 21324139; PubMed Central PMCID: PMCPMC3055815.

---

## [Decision Letter · Decision Letter 1]

31 Aug 2020

Objective quantitative multidetector computed tomography assessments in patients with combined pulmonary fibrosis with emphysema: relationship with pulmonary function and clinical events

PONE-D-20-02957R1

Dear Dr. Suzuki,

We’re pleased to inform you that your manuscript has been judged scientifically suitable for publication and will be formally accepted for publication once it meets all outstanding technical requirements.

Kind regards,

Y. Peter Di, Ph.D.

Academic Editor

PLOS ONE

Additional Editor Comments (optional):

Reviewers' comments:

Reviewer's Responses to Questions

**Comments to the Author**

1. If the authors have adequately addressed your comments raised in a previous round of review and you feel that this manuscript is now acceptable for publication, you may indicate that here to bypass the “Comments to the Author” section, enter your conflict of interest statement in the “Confidential to Editor” section, and submit your "Accept" recommendation.

Reviewer #1: All comments have been addressed

Reviewer #3: All comments have been addressed

2. Is the manuscript technically sound, and do the data support the conclusions?

Reviewer #1: (No Response)

Reviewer #3: Yes

3. Has the statistical analysis been performed appropriately and rigorously? 

Reviewer #1: (No Response)

Reviewer #3: Yes

4. Have the authors made all data underlying the findings in their manuscript fully available?

Reviewer #1: (No Response)

Reviewer #3: Yes

5. Is the manuscript presented in an intelligible fashion and written in standard English?

Reviewer #1: (No Response)

Reviewer #3: Yes

6. Review Comments to the Author

Reviewer #1: (No Response)

Reviewer #3: The authors have comprehensively addressed all of my thoughts and suggestions with regards to the manuscript.

7. PLOS authors have the option to publish the peer review history of their article (what does this mean?). If published, this will include your full peer review and any attached files.

Reviewer #1: No

Reviewer #3: No

---

## [Editor Report · Acceptance letter]

3 Sep 2020

PONE-D-20-02957R1 

Objective quantitative multidetector computed tomography assessments in patients with combined pulmonary fibrosis with emphysema: relationship with pulmonary function and clinical events 

Dear Dr. Suzuki:

I'm pleased to inform you that your manuscript has been deemed suitable for publication in PLOS ONE. Congratulations! Your manuscript is now with our production department. 

Kind regards, 

on behalf of

Dr. Y. Peter Di 

Academic Editor

PLOS ONE